# Redesigning Multimodal Interaction: Adaptive Signal Processing and Cross-Modal Interaction for Hands-Free Computer Interaction

**DOI:** 10.3390/s25175411

**Published:** 2025-09-02

**Authors:** Bui Hong Quan, Nguyen Dinh Tuan Anh, Hoang Van Phi, Bui Trung Thanh

**Affiliations:** 1Faculty of Information Technology, VNU-University of Engineering and Technology (VNU-UET), Hanoi 10000, Vietnam; 22028016@vnu.edu.vn (B.H.Q.); 22028136@vnu.edu.vn (N.D.T.A.); 22028167@vnu.edu.vn (H.V.P.); 2Faculty of Mechanical Engineering, Hung Yen University of Technology and Education, Hungyen 16000, Vietnam

**Keywords:** human-computer interaction, hands-free interaction, vision/camera-based sensors, adaptive signal processing, multimodal interaction, assistive technology

## Abstract

Hands-free computer interaction is a key topic in assistive technology, with camera-based and voice-based systems being the most common methods. Recent camera-based solutions leverage facial expressions or head movements to simulate mouse clicks or key presses, while voice-based systems enable control via speech commands, wake-word detection, and vocal gestures. However, existing systems often suffer from limitations in responsiveness and accuracy, especially under real-world conditions. In this paper, we present 3-Modal Human-Computer Interaction (3M-HCI), a novel interaction system that dynamically integrates facial, vocal, and eye-based inputs through a new signal processing pipeline and a cross-modal coordination mechanism. This approach not only enhances recognition accuracy but also reduces interaction latency. Experimental results demonstrate that 3M-HCI outperforms several recent hands-free interaction solutions in both speed and precision, highlighting its potential as a robust assistive interface.

## 1. Introduction

Advances in artificial intelligence (AI) are rapidly improving machines’ ability to process and understand visual data [1]. In addition, AI also promotes progress in fields such as robotics and education [2]. These developments are creating new opportunities to support accessible human-computer interaction, especially for individuals with disabilities [3]. One of the main objectives of assistive technology is supporting individuals with upper-limb impairments [4,5,6,7,8]. This group includes people with limb amputations, neuromuscular disorders such as amyotrophic lateral sclerosis (ALS), cerebral palsy, muscular dystrophy, spinal cord injuries, and congenital limb differences, all of which can significantly hinder the use of conventional input devices such as the mouse or keyboard.

According to [9], as of 2019, there were approximately 552.45 million people living with traumatic amputations. Additionally, nearly 33,000 people in the U.S. are currently living with ALS, and that number is projected to reach 36,000 by 2030 [10]. These conditions make it difficult or nearly impossible for individuals to use a computer mouse. However, the ability to move the head and eyes is often retained, even in individuals living with severe disabilities; therefore, computer interfaces based on head or eye movement are commonly employed as alternative input methods.

Recognizing this issue, many researchers have proposed solutions to support people with disabilities in accessing and interacting with computers effectively. Existing approaches can be broadly divided into two categories. Sensor-based systems (e.g., tilt sensors, accelerometers, gyroscopes) provide fast and accurate control but are costly, require technical setup, and may cause discomfort when worn for long periods [11,12,13,14,15,16]. Camera-based systems using standard RGB webcams are more affordable and convenient, ranging from early vision-based techniques [17,18,19,20] (template matching, color tracking) to more recent deep learning methods for head pose estimation, facial expression recognition, and gaze estimation [4,6,8,21,22,23,24,25,26].

While various camera-based approaches have been proposed to help individuals with disabilities control computers, most still suffer from latency caused by noisy input and inefficient filtering. Recent systems also rely on limited modalities and a small set of facial gestures (for example, smiling, mouth opening, eyebrow raising), which reduces flexibility and may interfere with precise cursor control due to involuntary movements. Moreover, input modalities are often treated in isolation, with little use of cross-modal integration. A promising direction is to exploit contextual cues, for example, validating voice commands with mouth opening detection, but this remains underexplored.

To address these challenges, we propose a novel 3-Modal Human-Computer Interaction (3M-HCI) system that integrates three complementary input modalities, namely head movement with facial expressions, voice commands, and eye gaze. Our contributions are threefold:

We introduce an adaptive filtering mechanism to suppress signal noise while maintaining low-latency responsiveness.We redefine the mapping strategy from input signals to cursor movements for more precise control.We incorporate cross-modal coordination to improve reliability and reduce false activations.

To evaluate the system, two types of tests were conducted: (a) functional tests under various technical and environmental conditions; (b) user evaluations to assess usability, responsiveness, and perceived effectiveness.

The remainder of this paper is organized as follows. Section 2 reviews related work on sensor-based and camera-based approaches for hands-free computer interaction. Section 3 introduces the system architecture and input modalities. Section 4 outlines the materials used and the evaluation methodology. Section 5 presents experimental results and discussions. Finally, Section 6 concludes the paper and suggests future research.

## 2. Related Work

Research on alternative computer input methods for individuals with upper-limb disabilities has evolved over the past two decades, moving from wearable sensor-based devices to camera-based solutions, and more recently toward vision models enhanced by deep learning. These approaches can be broadly categorized into two main directions: sensor-based and camera-based.

### 2.1. Sensor-Based Approaches

Early studies primarily focused on wearable sensor devices to capture head or body movements. A head-controlled computer mouse using tilt sensors was developed for people with disabilities, where one sensor detects horizontal movement and another detects vertical movement, and a touch switch allows click actions [11]. Some studies [12,13] employed a dual-axis accelerometer to control the mouse. Another approach used a combination of a gyro sensor and an optical sensor to perform clicking actions [14]. Additionally, a system for controlling a computer mouse using a camera and software mounted on a cap worn by the user has been developed [15]. Other studies [16] combined eye tracking and head gestures, using a light source mounted on the user. These methods have demonstrated particularly fast and accurate results. However, they all rely on sensor devices. This increases deployment costs, making it difficult for users in low-income areas or those without access to advanced technology. Furthermore, installing and calibrating sensors often requires a certain level of technical skill, which not all users may possess. Additionally, wearing sensor devices on the head for extended periods can cause discomfort, neck fatigue, or a sense of heaviness, negatively affecting the user experience.

### 2.2. Camera-Based Approaches

To reduce cost and improve accessibility, researchers gradually shifted to camera-based approaches, typically relying on standard built-in RGB webcams available on consumer devices. Early methods relied on template matching [17,18] or color-based segmentation [19,20] to track head or facial features. Interaction mechanisms included dwell clicking (cursor dwell time to simulate left-click) or eye blinks for right clicks. Some systems combined head movements with voice commands [4,21], while others integrated head orientation with eye blinks [22].

With the advent of deep learning, camera-based methods gained robustness and precision. Some studies directly mapped visual input to screen coordinates [23,24], while others extracted facial landmarks and expressions for interaction [6,8,25]. Despite these advances, most systems still relied on a limited set of gestures (e.g., smiling, eyebrow raising, mouth opening), restricting user flexibility and sometimes interfering with natural expressions. Another common limitation is latency, which often arises from noisy input signals and insufficient filtering mechanisms, leading to slower system responses and reduced usability in real-world scenarios. Moreover, input modalities were typically handled in isolation without cross-modal coordination. While some studies have employed multiple modalities [4,26], they typically process inputs independently without cross-modal coordination. These limitations motivate the need for a more flexible, responsive, and integrated interaction framework, which we address in this work.

## 3. System Architecture

The 3-Modal Human Computer Interaction (3M-HCI) system employs a callback-based architecture that fundamentally separates processing logic from user interface components, ensuring optimal performance, maintainability, and scalability. The core processing pipeline operates independently in separate threads, with results delivered to the GUI for display (Figure 1). To achieve maximum efficiency, the system implements a multi-threaded architecture within a single process, with mechanisms for safe thread coordination and termination. By using primarily I/O-bound operations, the system maintains significantly lower resource overhead compared to existing solutions.

Figure 1 illustrates the overall system architecture and its integrated modules. Within the camera thread, OpenCV is adopted as the standard library for capturing and processing input images. Implemented in optimized C/C++ with multi-core support, it delivers high performance and a comprehensive set of tools for efficient real-time image processing [27]. Additional libraries are incorporated for specific modules, which will be described in detail in the following sections. The cross-modal coordination mechanism enhances interaction among modules: the face processor validates input for the voice processor, while the voice processor can adjust profile settings and influence the behavior of the face processor. Both modules interface with the operating system through PyAutoGUI, which enables seamless control of mouse and keyboard actions.

Building upon this modular architecture, the main processing pipeline operates through synchronous and asynchronous callbacks as illustrated in Figure 2. The computer vision thread continuously captures frames from the camera input. Through the callback mechanism, each frame is delivered to the face processor module. The face processor provides two processing modes: the IMAGE mode processes each frame separately and synchronously, while the smooth mode handles frames asynchronously. Upon successful processing, the extracted facial landmarks and blendshape data are forwarded through the pipeline to compute and execute mouse movements andkeyboard actions.

The voice processor functions as an independent module running in its own dedicated thread. It captures the microphone input and processes it through its recognition engine to identify pre-configured voice commands. To enhance user experiences and prevent false activations from external audio sources, the system incorporates an intelligent gating mechanism that cross-references facial expression data from the face processor module before executing voice commands.

This architecture demonstrates significant optimization advantages compared with existing solutions such as Google Project GameFace [6], which utilizes a busy-waiting pipeline that tightly couples GUI and processing components. Google’s implementation continuously captures and processes images at extremely high frequencies (sub-millisecond intervals), resulting in substantial resource consumption and redundant frame processing. In contrast, our event-driven approach with controlled frame rates delivers superior resource efficiency while maintaining real-time responsiveness. The detailed implementation of each module is described in the subsequent sections.

### 3.1. Video Processing Module

#### 3.1.1. Face Processing

The face processing modules’ main function is to detect and track the user’s face and return facial landmarks, eye gaze, and expressions, which makes them the most critical component in the entire system pipeline. The face processing algorithm must satisfy several requirements: it should be fast, deliver high accuracy, and ideally operate efficiently without requiring a GPU.

Previous systems [4] have typically used the Dlib library [28], the Haar Cascade algorithm [29], or custom-built CNNs for face detection and tracking [23,24,30]. Recent systems [6,8] have increasingly adopted MediaPipe [31] due to its superior performance and the wide range of built-in features it provides. We compare several lightweight face processing algorithms. All algorithms below were evaluated on a single laptop with a Ryzen 5 5500U CPU, 12 GB RAM, and AMD Radeon Vega 7 integrated graphics to ensure consistent performance comparisons. The result are presented in Table 1.

Based on the comparison results shown in the table above, we decided to use MediaPipe Face Landmarker as the core tool for developing our system. MediaPipe [31] provides 478 facial landmarks (Figure 3), including key regions such as the eyes, eyebrows, mouth, nose, and jawline, which are essential for precise facial expression analyses.

#### 3.1.2. Mapping Features to Mouse Coordinate

Camera-based head-mouse systems typically use landmarks such as the nose tip [4,8], forehead [6], or mouth [19] as anchors. However, landmark instability during facial expressions can introduce unwanted cursor drift. For example, raising an eyebrow, as seen in Google’s Project GameFace, can shift the forehead anchor, while actions such as nose sneer or performing gestures such as “mouse left” and “mouse right” can alter the nose tip position, leading to unintentional pointer movement. A simple way to mitigate this issue is to avoid using facial expressions that can distort the anchor points altogether. Therefore, our solution is to use the two inner eye corners (medial canthi) as anchor points instead. After thorough testing, we found that the medial canthi remained stable across various facial expressions. Therefore, we chose them as reliable reference points for pointer movements (Figure 4). The movement vector formed by tracking these two inner eye corners is then converted into mouse pointer movement signals.

Formally, let *P_L_* = (*X_L_*, *Y_L_*) and *P_R_* = (*X_R_*, *Y_R_*) be the coordinates of the left and right medial canthus, respectively. We computed the midpoint at frame t:(1)$$C_{t}=(\frac{X_{L}+X_{R}}{2},\;\frac{Y_{L}+Y_{R}}{2})$$

Finally, we have cursor displacement vector *V_t_*:(2)$$V_{t}\;=\;C_{t}-C_{t-1}=\;(\Delta_{x},\Delta_{y})$$

Displacement vector *V_t_* serves as the raw cursor moving signal in our system.

#### 3.1.3. Adaptive Movement Signal Filtering and Acceleration

With signals measured from sensors or cameras (since a camera itself is a type of sensor), there is always some noise present in the data. To eliminate this noise, we apply filtering techniques. Depending on the characteristics of the signal, different filtering methods can be used (i) for signals with significant “salt-and-pepper” noise, a median filter can be applied to remove outliers; (ii) for signals affected by Gaussian noise, a moving average, low-pass, or Gaussian filter may be applied.

In our specific case, the movement signals extracted from facial landmarks often contain Gaussian noise. This causes the cursor movement to appear jittery. To address this issue, some systems use a simple region-based technique, where a virtual window is overlaid on the user’s face and the cursor only moves when the tracked landmark crosses the boundary of this window [4]. Additionally, previous systems have applied low-pass filters as a basic solution to smooth the pointer motion [8,17]. A more optimized approach is to use a Hamming filter, which offers better frequency response characteristics [6].

Although these techniques help smooth the cursor movement, they can negatively impact the pointer speed and responsiveness. Conventional low-pass filters have a fixed cutoff frequency, which creates a trade-off between smoothness and responsiveness. If the cutoff is too low, the cursor becomes stable but sluggish; if it is too high, unwanted jitter may persist.

To increase responsiveness, we employed the 1€ filter [34], an adaptive filter that dynamically adjusts its cutoff frequency according to the signal’s speed. This allows the system to remain smooth during slow movements while still being responsive during rapid changes. Note that, instead of applying the 1€ filter directly to the *x* and *y* coordinates separately, we apply the 1€ filter to the cursor displacement magnitude to get the smoothing factor *α*. This approach avoids the issue of having different cutoff frequencies on each axis, which could cause asynchronous pointer behavior.

We define $Dt=\sqrt{{\Delta x}^{2}+{\Delta y}^{2}}$ to be the cursor displacement magnitude, using the 1€ filter to get the smoothing factor:(3)$$\hat{D_{t}}\;,\alpha_{t}=1€\_filter(D_{t})$$

Next, we calculate the filtered displacement vector, using adaptive smoothing factor $\alpha_{t}$:(4)$$\hat{V_{t}}=\alpha_{t}V_{t}+(1-\alpha_{t}){\hat{V}}_{t-1}$$

Finally, we apply a pointer acceleration function to improve the responsiveness and usability of the system. This allows small head movements to result in fine cursor control, while larger or faster movements produce quicker pointer displacement, enhancing both the precision and efficiency. Pointer acceleration is typically based on sigmoid functions. Previous studies [6,35,36,37] have demonstrated that sigmoid-based pointer acceleration achieves smoother transitions between precise and rapid movements, while avoiding excessive jitter or drift. It also improves the ergonomics and precision of the system. We adopted the following function:(5)$$G\left(x\right)=\frac{K}{1+e^{-slope*(x-offset)}}$$
where: *K* controls the maximum gain, determining how fast the pointer can move at high speeds; *slope* defines the steepness of the transition between low and high gain; larger values make the transition sharper; *offset* sets the inflection point on the input axis, i.e., the point at which the gain starts to increase significantly.

In our system, we set *K* = 1.2, *slope* = 0.1, and *offset* = 12. The resulting acceleration curve is illustrated in Figure 5, showing a smooth transition from low to high gain as the input speed increases.

#### 3.1.4. Mapping Facial Expressions to Mouse and Keyboard Actions

Early studies employed sensors [11,14] or dwell-click mechanisms [20], combined with a limited set of simple actions or expressions [4,19] to trigger mouse events. More recent approaches have shifted toward using more intuitive and user-friendly expressions such as smiling, raising eyebrows, or opening the mouth to enhance usability and reduce fatigue [4,6,8]. However, these systems typically utilize only a small number of facial expressions [4,8], as some expressions have been reported to be difficult to perform or maintain and may cause facial fatigue. In addition, certain expressions can be easily confused with one another, reducing the reliability of the input [6,25].

A straightforward way to address this issue is to use only easily recognizable and distinguishable facial expressions that are less likely to be confused with others. However, this approach inherently limits the number of distinct actions the system can support.

To overcome this limitation, we introduce a priority-based triggering mechanism, which favors less distinguishable expressions over more easily recognizable ones when multiple expressions are detected simultaneously. For instance, when a user smiles, the model may also detect mouth opening due to overlapping facial features. In such cases, our system prioritizes the smile over the mouth opening to ensure consistent and reliable input. By using this mechanism, our system supports a wider range of actions compared to existing systems (Table 2), while also reducing false positives and confusion.

Moreover, in our system, we also utilized directional eye gaze (left, right, up, down) as a form of expressive input, similar to facial expressions.

### 3.2. Voice Processing Module

The voice processing module utilizes command recognition to handle basic interactive instructions and accessibility controls. It serves as a complementary input method to the facial expression control system, providing users with multiple interaction modalities. The module should use a pretrained model, support offline functionality, and offer high processing speed. Recent advances in speech recognition, such as Whisper [38] and Vosk [39], provide powerful accuracy and flexibility across languages. Nevertheless, these systems involve complex implementation and their real-time voice command recognition performance is limited on lightweight devices.

Since voice command is not the primary focus of our system, we prioritize deployment efficiency and adopt Microsoft’s native Speech API (SAPI5) [40] as a standard solution that satisfies the required criteria. Integrated through the Dragonfly library, SAPI5 delivers consistent performance with low latency and minimal hardware requirements, while executing all processing locally. This ensures that the feature operates without requiring an Internet connection, thereby enhancing privacy and system reliability.

The voice processor module creates a self-contained, thread-safe service that listens for user-defined voice commands (Figure 2). When a command is recognized by the SAPI5 engine, it executes the corresponding keyboard or mouse action via the PyAutoGUI. Furthermore, it interfaces with other modules within the application, allowing voice commands to modify their behavior. This multimodal approach also enhances the user experience. For instance, by implementing a check to determine whether the user’s mouth is open during command recognition, the system can avoid misinterpreting external ambient sounds as commands, leading to more reliable activation. Voice commands can also be used to dynamically adjust the mouse movement speed, enabling users to fine-tune their control in real time without relying on manual input. Furthermore, the voice command module can be flexibly configured to trigger interactive actions within the system, enhancing the overall user convenience.

## 4. Materials and Methods

This section outlines the development environment and the methodology used to evaluate the performance and usability of our multimodal interaction system. The evaluation comprises two types of tests. The first is an experimental test designed to examine system stability and responsiveness under various environmental and hardware conditions, including different CPU generations, operating systems, lighting environments, and background noise. The purpose is to determine the minimum requirements necessary for smooth operation. The second test involves task-based usability testing, in which users are asked to perform a series of predefined actions such as cursor movement and target selection. Objective performance metrics, including latency, accuracy, jitterness, and task completion time, are recorded and compared across systems. In addition, we conducted a short survey to collect subjective feedback from participants who had experienced all three systems, with the results presented in Section 5. This approach allows for a comprehensive analysis of both the technical efficiency and user experience.

### 4.1. Development Platform

We used two different laptops during the development process to ensure stable software performance under varying hardware conditions. The first laptop was equipped with a Ryzen 5 5500U CPU, 12 GB RAM, AMD Radeon Vega 7 integrated graphics, a built-in 720 p 30 fps camera, and an integrated microphone. The second laptop featured an Intel(R) Core(TM) i7-13650HX CPU, 16 GB RAM, a dedicated NVIDIA GeForce RTX 4060 GPU with 8 GB GDDR6 VRAM, a built-in 720 p 15 fps camera, and an integrated microphone.

The system was developed through iterative prototyping, combined with regular internal testing and feedback from university instructors and experts with experience in assistive technologies. This feedback loop allowed us to continuously improve the system while keeping it accessible and practical. To implement our application, we chose Python as the primary programming language due to its extensive ecosystem, cross-platform compatibility, and active developer community. Python also simplifies rapid prototyping and integration with computer vision and audio processing tools, which are central to our system. The key Python libraries utilized included: (i) OpenCV for video capture and preprocessing, (ii) Mediapipe for extracting facial landmarks and facial expression analyses, (iii) Customtkinter for building modern and customizable graphical user interfaces (GUIs), (iv) Dragonfly for a voice control framework that maps spoken commands to computer actions, (v) PyAutoGUI for accessing the mouse and keyboard functionalities, and (vi) NumPy for efficient numerical computations.

The main interface of the application is illustrated in Figure 6. Detailed information about the project and the source code of the project are available at: https://github.com/ndtuananh04/3-Modal-Human-Computer-Interaction (accessed on 16 July 2025).

### 4.2. Testing Methodology

Following the evaluation of the methodology proposed in [4], we assessed the minimum operating requirements of our system under various environmental and technical conditions, such as lighting, background noise, and hardware configurations, as follows: (a) different environmental lighting conditions; (b) more than one face detected by the camera; (c) background noise; (d) different hardware and software features of the computer. In addition to these tests, we also conducted task-based usability testing with existing systems, in order to highlight the effectiveness of the improvements proposed in our study: (a) jitterness; (b) responsiveness (task completion time); (c) accuracy.

Before performing the tasks, participants were given time to freely explore and adjust the mouse control system to ensure maximum comfort. They were instructed to select two facial expressions of their choice and assign them to left and right click actions, based on what they found most intuitive and easy to perform. Once these settings were configured, participants received clear instructions on how to interact with the testing application.

The first task involved a sequence-based interaction test, where users were required to move the cursor to predefined targets on the screen, perform either a left or right click as instructed, and proceed to the next target (Figure 7). This process continued until all targets were completed. The task was used to assess accuracy and responsiveness, based on metrics such as the completion time and cursor deviation.

The second task required users to keep their heads still for a fixed duration while the system was running. This allowed us to measure unintended cursor movement or drift, providing insight into the system’s stability when idle.

Finally, a short post-test survey was conducted to gather subjective feedback from users who had experienced all three systems (Table 3). All questions were evaluated on a numeric rating scale from 1 (very bad) to 10 (excellent). The results of this evaluation, as well as the code and configurations used in the testing application, are available in our public GitHub repository.

## 5. Results and Discussion

### 5.1. Robustness Testing Under Environmental and Hardware Variations

We evaluated the system’s functionality and performance under varied lighting, multiple faces, and different hardware and operating systems, as well as identify the minimum hardware and environmental requirements necessary for stable operation.

#### 5.1.1. Lighting Condition Test

To assess the robustness of 3M-HCI under real-world usage, we conducted experiments in four different lighting environments, with corresponding results shown in Figure 8a–d. In each scenario, we visualized the 147 facial landmarks used by MediaPipe Face Mesh to detect expressions, enabling a detailed qualitative assessment of detection stability under varying illumination conditions. From our experiments, we can conclude the following:Bright eEnvironment (Figure 8a,b): Whether in a brightly lit room or a dim room with high screen brightness, the system performed flawlessly. Facial landmarks were immediate and accurate. The mouse control operated smoothly without any jitter or delay. This represents the optimal environment for system usage.Dim room with medium screen brightness (Figure 8c): Under significantly darker conditions, where only moderate screen brightness was present, the system remained functional. Facial landmarks still worked, but occasional instability in mouse movement was observed. The system was still usable with minor degradation.Near-total darkness with low screen brightness (Figure 8d): In the most extreme case, with no external light and very low screen brightness, the system struggled. The Mediapipe framework could still detect the facial landmarks. However, the detection was inconsistent and unreliable. Landmarks often flickered or were lost entirely, making interaction with the system ineffective in this condition

From the experiments above, we conclude that the most critical factor for the system’s performance is the clarity of the captured facial image. While ambient lighting conditions have a limited impact on the overall outcome, ensuring a well-defined face is essential. Notably, MediaPipe demonstrated impressive robustness. It was able to detect facial landmarks even in extremely low-light scenarios where the human eye struggles to distinguish facial features.

#### 5.1.2. Multiple Faces in a Frame

This test evaluated Mediapipe’s behavior when multiple faces appear in the frame. When configured to detect only a single face, Mediapipe selects the one it detects with the highest confidence. In practice, this is often the face closest to the camera, which typically belongs to the user. However, the most confidently detected face is not always the intended user’s face, especially in dynamic or crowded environments. Our experiments show that when two faces are present in the frame, Mediapipe may occasionally select the one farther from the camera, which disrupts the system’s operation.

A simple strategy to address this issue is to enable Mediapipe’s multi-face detection mode. In this configuration, the system detects all visible faces and compares them to the face identified in the previous frame, selecting the one with the most consistent position or landmark pattern. While this improves the accuracy of user tracking in multi-face scenarios, it also introduces a higher computational load, which may reduce the real-time performance, particularly on lower-end devices. For this reason, we did not adopt this approach in our implementation, as it caused noticeable lag during runtime, making the interaction experience less smooth and responsive.

#### 5.1.3. Background Noise

In contrast to previous systems that relied solely on voice recognition, making them vulnerable to ambient noise and unintended speech, 3M-HCI integrates a mouth-open detection mechanism using facial landmarks. To evaluate its robustness, we conducted a test scenario where two people held a conversation near the system. While earlier studies [4] reported performance degradation due to microphone sensitivity and background noise, our method was unaffected. Since voice commands in our system are only executed when the user’s mouth has been recently detected as open, environmental noise or nearby conversations had no impact on command triggering. This approach significantly reduces false positives and enhances reliability in shared or noisy environments.

However, this feature relies on the system’s ability to consistently detect the user’s full face. If the face is partially occluded, out of frame, or poorly lit, the mouth-open detection may fail to activate, thereby preventing valid voice commands from being registered. Ensuring a clear and stable view of the user’s face is therefore essential for maintaining the robustness of this mechanism.

#### 5.1.4. Different Hardware and Software Features of the Computer

We evaluated the software performance across different machines and conducted a comparative analysis of three applications: 3M-HCI, Project GameFace, and CameraMouseAI. The results are summarized in Table 4 below.

Compared to other applications, our software runs efficiently and stably. Despite integrating voice commands, our system still consumes fewer computational resources than Google’s solution. CameraMouseAI exhibits lower computational cost than ours, likely because it is a simpler tool, supporting only 2–3 basic mouse actions.

Based on Table 4, we recommend the following minimum system requirements: Windows 10 or higher, a CPU equivalent to Intel Core i5-10th Gen or above, no dedicated GPU is required, a functional camera and microphone, and at least 8 GB of RAM. These are very lightweight requirements, making the system compatible with nearly all modern laptops.

### 5.2. Empirical Task-Based Test

We conducted a pilot user study involving eight participants to empirically evaluate the usability, responsiveness, and precision of the proposed 3M-HCI system in real-world interaction scenarios. All participants provided informed consent prior to the study. The objective of this test was to assess how effectively users could perform common cursor-based tasks using hands-free input, in comparison with traditional mouse input and two baseline systems: Project GameFace and CameraMouseAI. Each participant was asked to complete a set of target selection and click tasks under identical conditions across all systems. Key performance metrics, including pointer accuracy, latency, jitter, and number of successful clicks, were recorded. Additionally, a post-test survey was conducted to capture user satisfaction and perceived usability. The findings from this pilot study provide preliminary insights into the practical viability and comparative advantages of our system in assistive computing contexts.

#### 5.2.1. System Accuracy

We measured the system’s accuracy by calculating the deviation of four systems (CameraMouseAI, Project GameFace (Googl)e, our method (3M-HCI), and normal mouse) from the optimal path (straight path) to the target, along with the number of clicks required to complete the task. The results are presented in Figure 9 and Figure 10 below.

Overall, our system demonstrated higher accuracy compared to CameraMouseAI and Project GameFace. Overshooting was not observed in our system, in contrast to CameraMouseAI, where it occurred frequently during cursor movement. Regarding the clicking mechanism, both CameraMouseAI and Project Gameface experienced unintended cursor movement during facial expression activation, leading to incorrect clicks. Our system did not encounter this issue due to a more optimal selection of anchor points.

#### 5.2.2. System Responsiveness

We also measured the time taken to complete the first task as an indicator of system responsiveness. The result is shown in Figure 11.

As shown in the figure, our method consistently outperformed both CameraMouseAI and Project GameFace in terms of movement latency. Across all target transitions (e.g., 1 → 2, 2 → 3, etc.), our system maintained latency values of around 2 to 3 s, with minimal fluctuation. In contrast, CameraMouseAI exhibited the highest latency, with several spikes exceeding 6 s—most notably in the 5 → 6 transition. Project GameFace also showed relatively high latency, particularly during transitions 4 → 5 and 6 → 7.

Moreover, the total average latency of our system (green dashed line) is clearly lower than that of CameraMouseAI (blue dashed line) and Project GameFace (orange dashed line), indicating faster and more consistent performance. While traditional mouse input (red line) remained the fastest as expected, our method showed a strong balance between speed and usability, especially considering its hands-free nature.

#### 5.2.3. System Jitterness

As illustrated in Figure 12, our method achieved significantly lower jitter deviation compared to other hands-free systems. Specifically, the CameraMouseAI showed the highest instability with a jitter deviation of over 120 pixels, followed by Project GameFace with approximately 80 pixels. In contrast, our method maintained a low deviation of under 10 pixels, indicating stable and precise cursor control. As expected, traditional mouse usage yielded the lowest jitter rate, serving as a performance baseline.

This finding highlights the importance of incorporating adaptive filter algorithms and activation mechanisms in hands-free systems. By minimizing cursor jitter, our approach improves not only task precision but also user comfort and trust, which are critical for sustained use, especially among individuals with motor impairments.

### 5.3. Survey Results

The survey results provide insights into the subjective usability, responsiveness, and comfort of our system compared to existing solutions, including CameraMouseAI, Project GameFace, and traditional mouse control. We present the results below in Table 5.

The results in the table indicate that our proposed 3M-HCI system achieved high subjective ratings across all categories. Notably, it received an average score of 8.25 ± 1.09 for the responsiveness of left and right mouse clicks and 8.88 ± 0.6 for overall cursor responsiveness, comparable to the traditional mouse and outperforming both CameraMouseAI and Project GameFace.

In terms of ease of use, participants reported that our system required relatively little time to master (7.25 ± 1.71), and clicking actions were less difficult (7 ± 1.87) compared to other hands-free systems. One contributing factor is that our system allows users to choose from multiple facial expressions for click activation. Since different users may find certain expressions easier or more natural to perform, this flexibility improves comfort and accessibility. Additionally, our system maintains cursor stability during facial expression recognition, avoiding unintended cursor jumps—an issue observed in other systems such as CameraMouseAI and Project GameFace. Precision and directional control (both vertical and horizontal) were also rated higher than for alternatives, indicating more stable and accurate performance.

Fatigue levels while using the system were moderate (7.25 ± 1.79), significantly better than CameraMouseAI (2.62 ± 1.93) and slightly better than Project GameFace (7 ± 1.87), showing the ergonomic advantages of our approach. Importantly, our system received the highest rating (7.85 ± 2.71) in terms of perceived applicability for people with disabilities, suggesting strong user confidence in its real-world assistive potential.

### 5.4. Limitations

Despite the overall positive results, several limitations were identified based on user feedback during testing. One notable issue was the lack of intuitive parameter tuning, as users found it difficult to adjust the minCutoff and beta values of the 1-Euro filter. This difficulty stems from the technical definitions of minCutoff and beta, making it challenging to grasp their impact on the filtering process. Consequently, this limitation significantly hinders accessibility and the ability to optimize the filter’s performance. This complexity contrasts with earlier approaches, where the low-pass filter coefficient alpha was abstracted into a single, more intuitive parameter for smoothing or responsiveness. Another concern was microphone instability on low-end devices, where users reported delays in microphone initialization or instances where speech was not recognized, particularly during system startup or under constrained hardware conditions.

In this paper, the voice command component was not explored in depth. We selected a lightweight and general-purpose voice recognition module to ensure broad compatibility and minimal computational overhead. The primary design criteria were simplicity, low latency, and ease of integration. However, more advanced alternatives could be considered. For instance, integrating modern speech recognition frameworks such as OpenAI Whisper [38] may offer improved robustness, especially in noisy environments. In addition, exploring non-speech voice command systems [41] could further enhance the responsiveness, which is particularly beneficial for gamers. Additionally, our system currently underutilizes eye inputs. While eye direction is used as a gesture trigger, the system does not yet leverage richer gaze data for pointer control or attention estimation. Enhancing the eye-tracking integration could significantly improve the precision and interaction depth, especially for users with limited facial mobility.

## 6. Conclusions

This paper presents 3M-HCI, a novel, low-cost, and hands-free human-computer interaction system that integrates facial expressions, head movements, eye gaze, and voice commands through a unified processing pipeline. The central contribution of 3M-HCI lies in its unified processing architecture, which integrates three key components: (1) a cross-modal coordination mechanism that synchronizes facial, vocal, and eye-based inputs to enhance reliability and reduce false triggers; (2) an adaptive signal filtering method that suppresses input noise while maintaining low-latency responsiveness; (3) a refined input-to-cursor mapping strategy that improves control accuracy and minimizes jitter.

Our system enhances stability by reducing cursor jitter eightfold to under 10 pixels, while maintaining better responsiveness and accuracy than recent systems. Our system also performs well under challenging conditions, such as background noise, poor lighting, and low-end hardware configurations, compared to previous systems. This superiority was also reflected in direct user feedback, which awarded our system high scores for responsiveness (8.88/10), precision (8.37/10), and usability (7.85/10).

Although experimental results demonstrate that 3M-HCI outperforms recent baseline models in both accuracy and responsiveness, the system still requires further refinement. Future work will focus on three key directions. First, we aim to improve the usability by simplifying the tuning of the One Euro Filter through a real-time interface that abstracts low-level parameters such as the minCutoff and beta. Second, we plan to explore richer use of eye inputs, extending beyond simple gesture triggers toward gaze-based pointer control and attention estimation. Finally, we intend to investigate more versatile voice modules that provide greater robustness in noisy environments and broader adaptability across devices. Together, these directions will enhance the adaptability, precision, and inclusiveness of the 3M-HCI system.

## Figures and Tables

**Figure 1 sensors-25-05411-f001:**
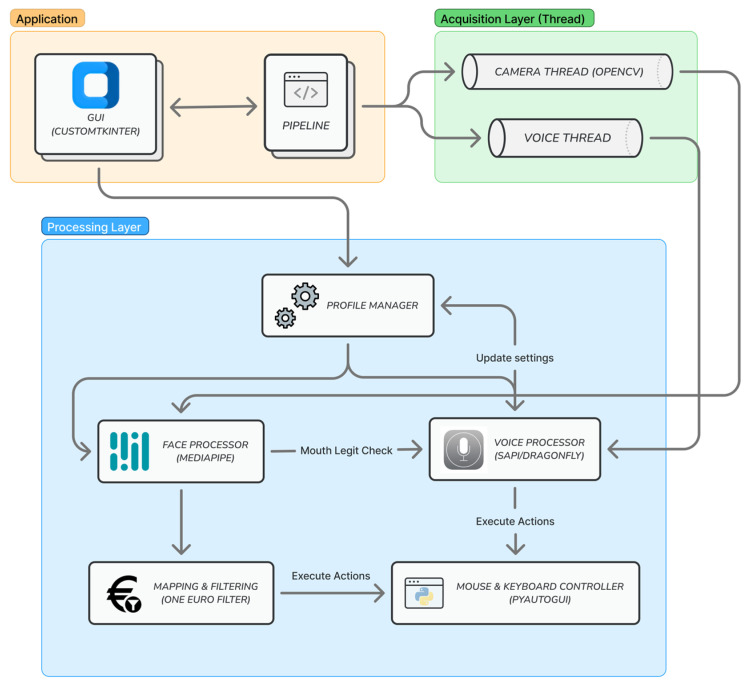
Overview of system architecture and processing modules.

**Figure 2 sensors-25-05411-f002:**
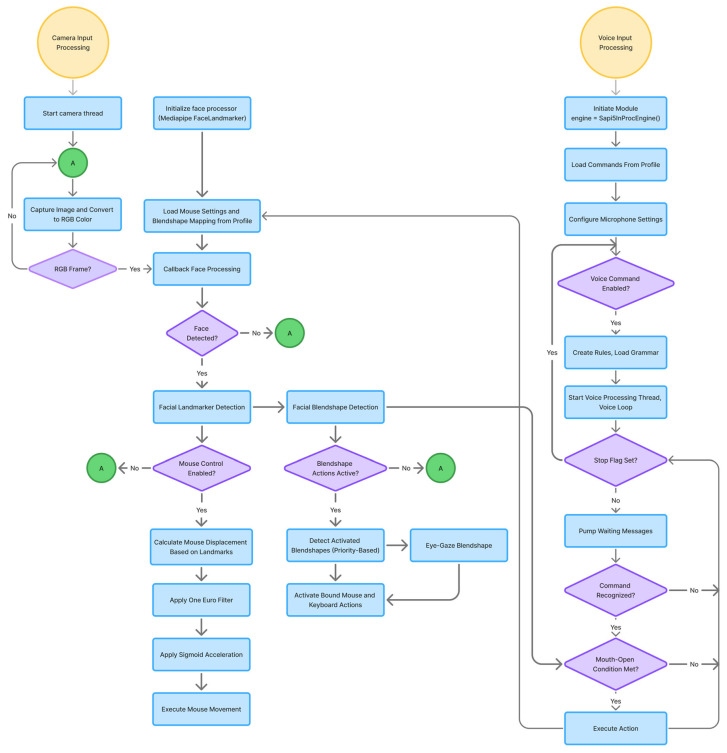
Multimodal input processing pipeline.

**Figure 3 sensors-25-05411-f003:**
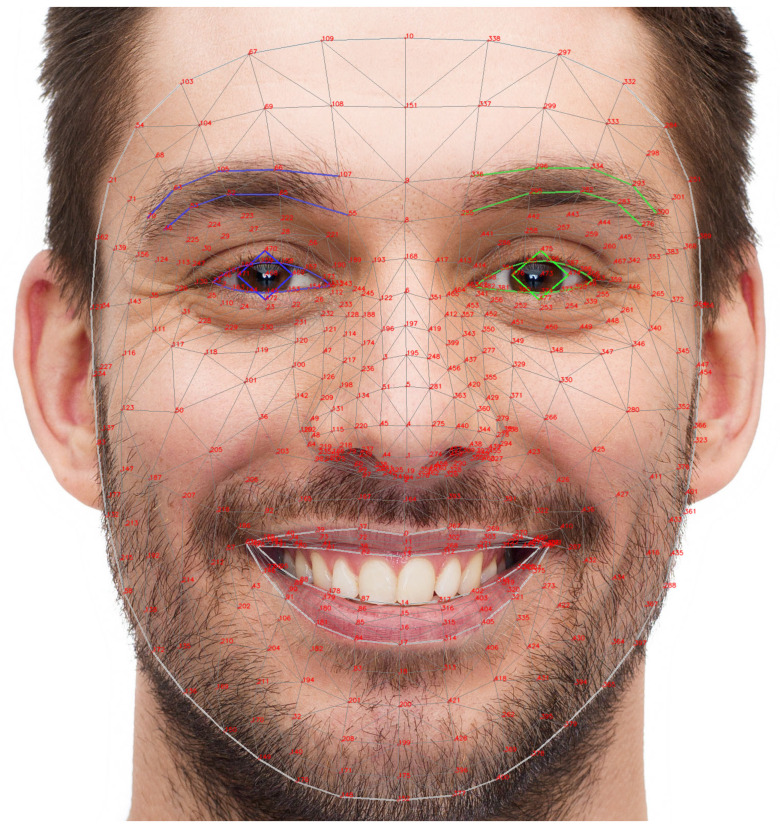
Mediapipe 478 landmarks. Each landmark point corresponds to a specific part of the face.

**Figure 4 sensors-25-05411-f004:**
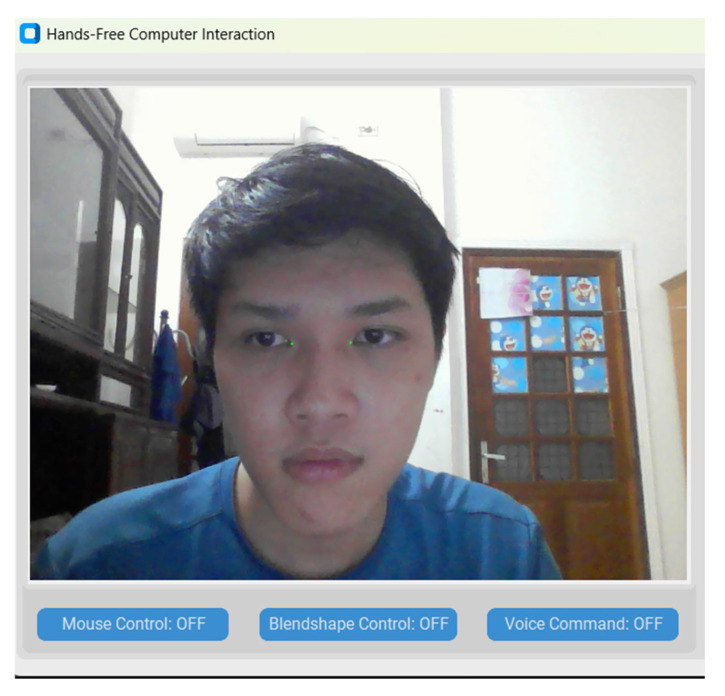
Two inner eye corners (p133 and p362 in Mediapipe).

**Figure 5 sensors-25-05411-f005:**
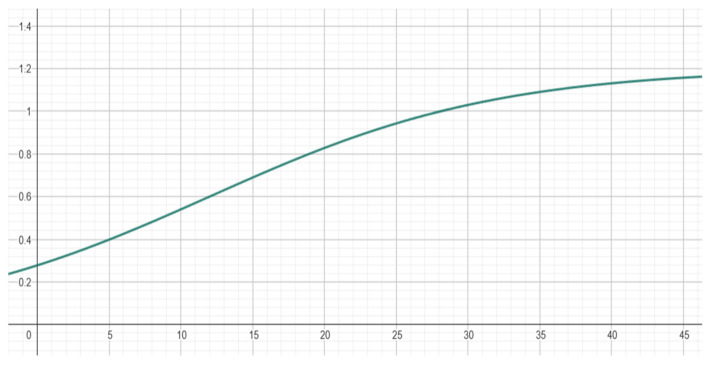
Acceleration curve.

**Figure 6 sensors-25-05411-f006:**
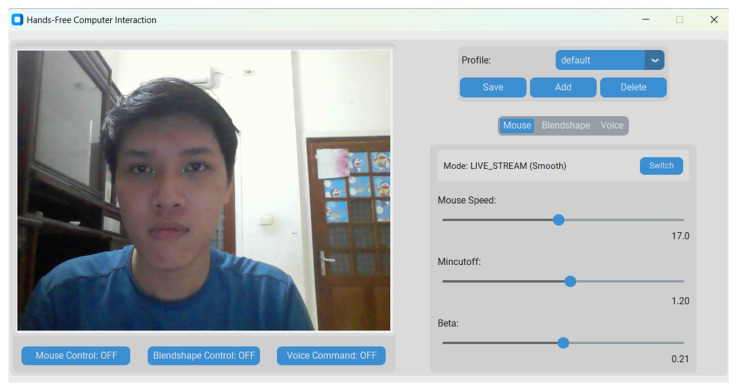
The 3M-HCI graphical user interfaces built with Customtkinter.

**Figure 7 sensors-25-05411-f007:**
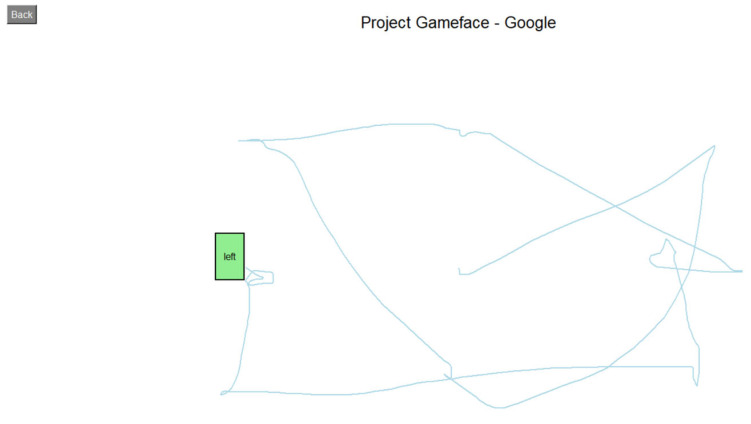
Moving and clicking tasks.

**Figure 8 sensors-25-05411-f008:**
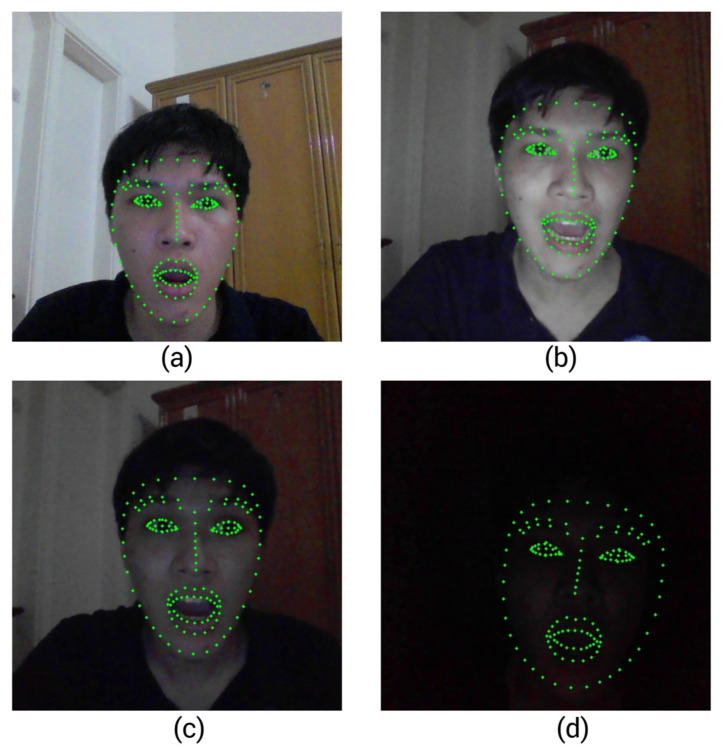
Mediapipe facial landmarks detection in different light conditions. (**a**,**b**) Bright environment. (**c**) Dim environment. (**d**) Dark environment.

**Figure 9 sensors-25-05411-f009:**
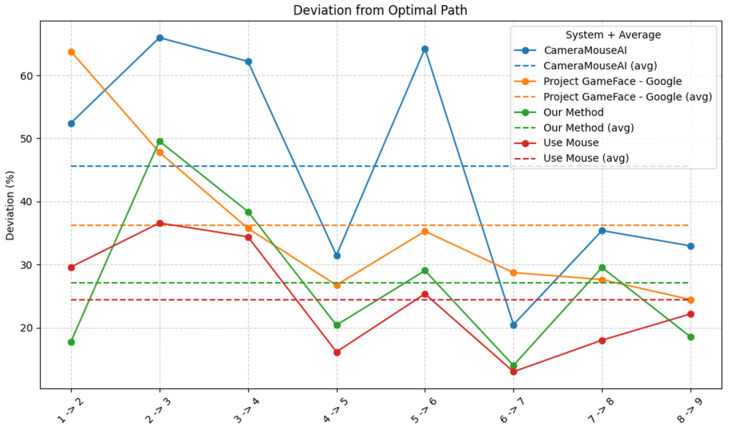
Deviations from optimal path.

**Figure 10 sensors-25-05411-f010:**
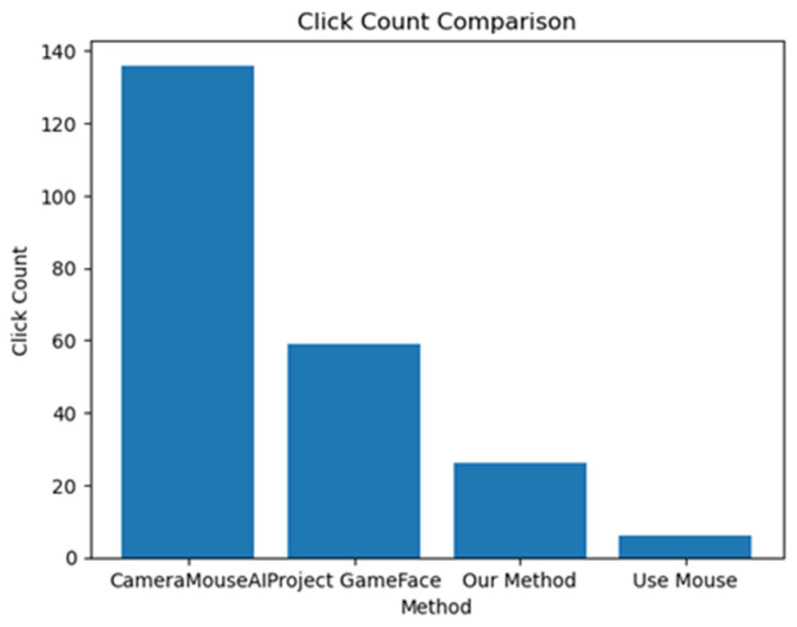
Number of clicks.

**Figure 11 sensors-25-05411-f011:**
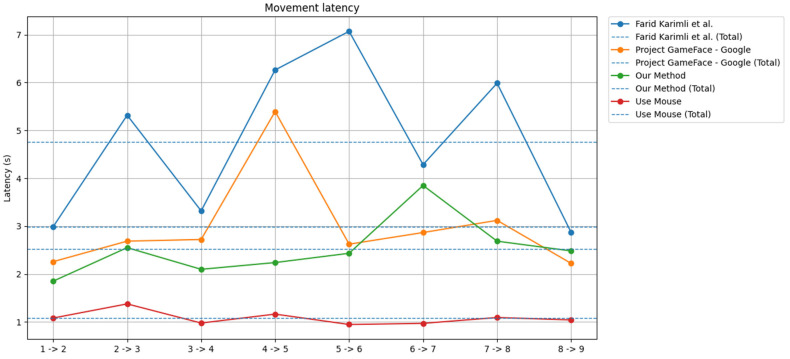
Movement latency.

**Figure 12 sensors-25-05411-f012:**
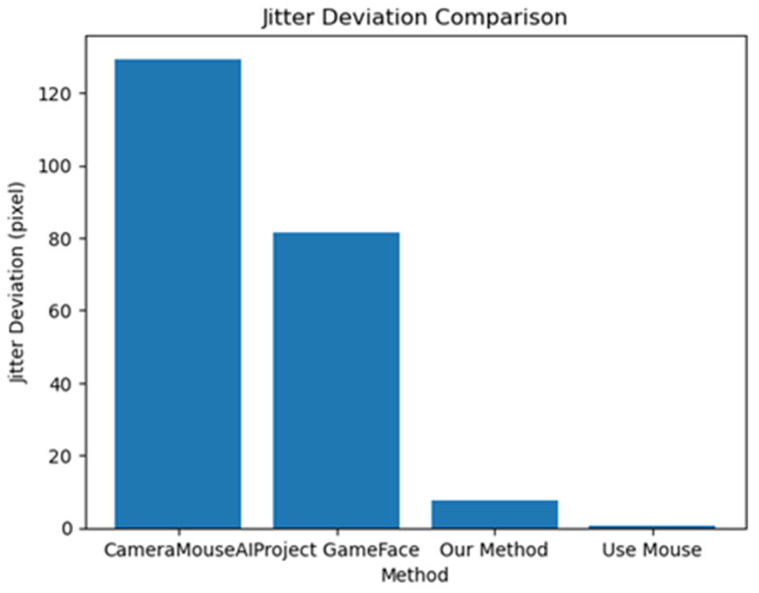
Jitter deviation of different systems.

**Table 1 sensors-25-05411-t001:** Face processing algorithm comparison.

Algorithm/Library	Number of Landmarks	Detection Time (second)	Detection Rate ^1^	Facial Expression	Iris Tracking
Dlib [28]	68	0.036	0.78	Some facial expressions can be computed manually ^2^	No
Mediapipe [31]	478	0.0037	1	52 built-in blendshapes	Yes
Haar Cascade [29]	0	0.0056	0.65	No	No
MTCNN [32]	5	0.215	1	No	No
YuNet [33]	5	0.026	0.93	No	No

^1^ Percentage of faces correctly detected among all images containing a face. ^2^ Following [4], we can calculate the mouth being open, for example.

**Table 2 sensors-25-05411-t002:** Comparison of our system with facial expression control interfaces.

System	#Facial Expression	Mouse Control	Keyboard Control	System Control	Triggering Mechanism
EMKEY [5]	1	-	-	x	Predefined Threshold
CameraMouseAI [8]	2	x	-	-	User-Defined Threshold
Project GameFace [6]	8	x	x	-	User-Defined Threshold
Zelinskyi et al. [34]	8	x	-	-	Predefined Threshold
3M-HCI (Ours)	13	x	x	x	User-Defined Threshold with Priority

**Table 3 sensors-25-05411-t003:** List of survey questions.

Question	Description
Q1	Does it take a lot of time to master the application?
Q2	Is the response of left/right mouse clicks fast?
Q3	Is the cursor movement responsive?
Q4	Is it difficult to click the left/right mouse button?
Q5	Is it difficult to move the cursor precisely?
Q6	Is it difficult to move the cursor vertically?
Q7	Is it difficult to move the cursor horizontally?
Q8	Does moving the cursor cause fatigue?
Q9	Do you think this mouse system can be applied to people with disabilities?

**Table 4 sensors-25-05411-t004:** Software performance on different laptops.

Laptop	CPU	RAM	OS	Overall Performance	Computational Cost (3M-HCI)	Computational Cost [6]	Computational Cost [8]
Dell Inspiron 15 3530	Intel Core i7-1355U	16 GB	Windows 11	Excellent	12.7%	15.1%	26.7%
Dell XPS 13 9360	Intel Core i7-7660U	16 GB	Windows 10	OK. The microphone takes time to boot	49.9%	60.3%	Unable to run
Dell G15 5530	Intel Core i7-13650HX	16 GB	Windows 11	Excellent	10.7%	23%	8.5%
Lenovo ThinkPad T480	Intel Core i5-8350U	8 GB	Windows 11	OK. The program is a bit laggy	52.8%	57.7%	31.2%
Dell Precision 7510	Intel Core i7-6820HQ	16 GB	Windows 10	Excellent	37.9%	41.5%	18.8%
MSI GF63 Thin 11UD	Intel Core i7-11800H	16 GB	Windows 11	OK. The microphone takes time to boot	19.7%	28.3%	Unable to run
Dell Inspiron 16 5620	Intel Core i5-1240P	16 GB	Windows 11	Excellent	6.1%	20.56%	16.2%
HP Laptop 16-d0xxx	Intel Core i5-11400H	8 GB	Windows 11	Excellent	24.7%	19.1%	11.05%
ASUS TUF Gaming F15	Intel Core i5-10300H	16 GB	Windows 11	Some commands cannot be recognized.	41.88%	47.37%	23.4%
MSI Modern 15	AMD Ryzen 5 5500U	12 GB	Windows 11	Voice command runs badly.	34.6%	47.4%	17.2%

**Table 5 sensors-25-05411-t005:** Survey results.

Question	Description	CameraMouseAI	Project GameFace	3M-HCI (Ours)	Mouse
Q1	Does it take a lot of time to master the application?	3.5 ± 1.87	6.75 ± 1.98	7.25 ± 1.71	10.0 ± 0.0
Q2	Is the response of left/right mouse click fast?	3.5 ± 2.35	7.5 ± 1.58	8.25 ± 1.09	10.0 ± 0.0
Q3	Is the cursor movement responsive?	5.38 ± 2.06	6.62 ± 1.11	8.88 ± 0.6	10.0 ± 0.0
Q4	Is it difficult to click the left/right mouse button?	4 ± 2.55	6.25 ± 1.56	7 ± 1.87	10.0 ± 0.0
Q5	Is it difficult to move the cursor precisely?	3.5 ± 2.45	6.87 ± 1.17	8.37 ± 0.7	10.0 ± 0.0
Q6	Is it difficult to move the cursor vertically?	4.5 ± 2.18	7.5 ± 1.41	8.37 ± 0.86	10.0 ± 0.0
Q7	Is it difficult to move the cursor horizontally?	4.5 ± 2.18	7.5 ± 1.41	8.37 ± 0.86	10.0 ± 0.0
Q8	Does moving the cursor cause fatigue?	2.62 ± 1.93	7 ± 1.87	7.25 ± 1.79	9.87 ± 0.33
Q9	Do you think this mouse system can be applied for people with disabilities?	4 ± 3.53	7.12 ± 2.52	7.85 ± 2.71	7.75 ± 3.9

## Data Availability

The data supporting the findings of this study are available from the corresponding author upon request. All source codes of the applications are available in this public repository: https://github.com/ndtuananh04/3-Modal-Human-Computer-Interaction (accessed on 20 August 2025).

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
