# Peer review of "Redesigning Multimodal Interaction: Adaptive Signal Processing and Cross-Modal Interaction for Hands-Free Computer Interaction"

_sensors, 2025, doi:10.3390/s25175411_

Round 1
Reviewer 1 Report
Comments and Suggestions for Authors
- The abstract is well written.
- The introduction is clearly articulated and well-structured, effectively setting the stage for the study. It provides a concise overview of the topic, highlights its significance, and outlines the main objectives. The flow of ideas is logical, making it easy for the reader to understand the context and rationale behind the research. It is concise and precise. But it contains improperly formatted citations, for example, "Pereira et al. [15]" the references are inconsistent: sometimes only a number is used, other times the author's name appears. The citation style should be uniform throughout the text.
- I couldn’t find a dedicated section for the literature review. Ideally, the paper should first provide a structured overview of the historical background and the current state of research on Human-Computer Interaction. This would help lay the foundation for understanding the context and relevance of the study.
- Please check the correct reference format in the template. Sometimes you wrote "Pereira et al. [15]," “Lin et al” while other times you used only the number. Ensure consistency throughout the paper.
- I couldn’t find a dedicated Related Work section. Including such a section is essential to position the study within existing research, highlight previous contributions in the field, and justify the novelty of the current work even if you specify some of them in the introduction.
- In Section 2.1.3 Adaptive Movement Signal Filtering and Acceleration, when discussing the elimination of noise from captured data, it would be helpful to provide further clarification. For instance, in the case of Gaussian noise, you should specify which linear filters are appropriate,such as moving average, low-pass, or Gaussian filters. Additionally, it is important to mention that the optimal noise removal method depends on the specific type of noise affecting the image you captured.
- There are several ways to measure cursor movement distance, so why are you using only cursor displacement magnitude when other methods are also available? In line 218.
- The results section is well described taking into account all lighting condition tests.
- The conclusion is well written; however, please note that conclusion sections typically should not include citations. Additionally, it is important to incorporate the results and the accuracy achieved by the system.
- The paper should undergo proofreading to eliminate grammatical and typographical errors.

Author Response
Comments 1: The abstract is well written.
Thank you for your positive comment.
Comments 2: The introduction is clearly articulated and well-structured, effectively setting the stage for the study. It provides a concise overview of the topic, highlights its significance, and outlines the main objectives. The flow of ideas is logical, making it easy for the reader to understand the context and rationale behind the research. It is concise and precise. But it contains improperly formatted citations, for example, "Pereira et al. [15]" the references are inconsistent: sometimes only a number is used, other times the author's name appears. The citation style should be uniform throughout the text.
We have carefully revised the manuscript to ensure that all citations are now formatted consistently.
Comments 3: I couldn’t find a dedicated section for the literature review. Ideally, the paper should first provide a structured overview of the historical background and the current state of research on Human-Computer Interaction. This would help lay the foundation for understanding the context and relevance of the study.
The former Introduction has been split into two distinct sections, Introduction and Related Work, for a clearer structure. Additionally, we have organized the Related Work section to offer a structured overview of the historical background and the current state of research in Human-Computer Interaction, making it easier for the reader to follow the context and relevance of our study.
Comments 4: Please check the correct reference format in the template. Sometimes you wrote "Pereira et al. [15]," “Lin et al” while other times you used only the number. Ensure consistency throughout the paper.
We have carefully revised the manuscript to ensure that all citations are now formatted consistently.
Comments 5: I couldn’t find a dedicated Related Work section. Including such a section is essential to position the study within existing research, highlight previous contributions in the field, and justify the novelty of the current work even if you specify some of them in the introduction.
Thank you for the suggestion. The former Introduction has been split into two distinct sections, Introduction and Related Work, for a clearer structure. Additionally, we have organized the Related Work section to offer a structured overview of the historical background and the current state of research in Human-Computer Interaction, making it easier for the reader to follow the context and relevance of our study.
Comments 6: In Section 2.1.3 Adaptive Movement Signal Filtering and Acceleration, when discussing the elimination of noise from captured data, it would be helpful to provide further clarification. For instance, in the case of Gaussian noise, you should specify which linear filters are appropriate,such as moving average, low-pass, or Gaussian filters. Additionally, it is important to mention that the optimal noise removal method depends on the specific type of noise affecting the image you captured.
We will add clarification regarding appropriate linear filters for Gaussian noise, such as moving average, low-pass, or Gaussian filters. Regarding the second point, although we acknowledge that there is no universally optimal filter for all types of noise, our system prioritizes responsiveness and a smooth user experience. Therefore, we chose the One Euro Filter, which is widely used in VR, browser (Chromium Blog: Smoothing out the scrolling experience in Chrome on Android) to smooth rapid movements while preserving low-latency responsiveness, enhancing overall user interaction.
Comments 7: There are several ways to measure cursor movement distance, so why are you using only cursor displacement magnitude when other methods are also available? In line 218.
We thank you for the valuable comment. In our work, we have two concepts, namely displacement vector Vt = (Δx, Δy) and magnitude . We use Vt as the motion signal (after filtering), while Dt represents the movement length and is used to determine the smoothing factor ? in the low-pass filter.
Many prior works directly apply filtering to the displacement vector along each axis:
This can be written as:
Or in Google’s Project Gameface, they use a Hamming-window FIR filter, i.e., a finite impulse response filter designed using a Hamming window to smooth the signal:
We initially applied the same method using the One Euro Filter along each axis separately:
However, filtering the X and Y components separately caused unsmooth cursor movement. This is because the two filters had different adaptive states: unlike the previous filters with fixed coefficients, the One Euro Filter’s effective α is dynamically adapted, leading to inconsistencies along each axis when moving diagonally. To address this, we applied the One Euro Filter to the displacement magnitude Dt instead of each individual component. We use the displacement magnitude Dt, instead of other possible metrics, such as |Δx| + |Δy|, because it provides a more intuitive and natural measure of overall movement length.
Comments 8: The results section is well described taking into account all lighting condition tests.
Thank you for your positive response.
Comments 9: The conclusion is well written; however, please note that conclusion sections typically should not include citations. Additionally, it is important to incorporate the results and the accuracy achieved by the system.
Thank you for the valuable feedback. We have revised the Conclusion section to include specific results and the achieved accuracy of our system. Regarding the citations that were originally in the Conclusion, we agree that it is not typically appropriate to include references there. These citations were mainly used to discuss current limitations and potential future directions. We have now moved them to the Limitations section, where they are more appropriate.
Comments 10: The paper should undergo proofreading to eliminate grammatical and typographical errors.
We have carefully proofread the manuscript and corrected grammatical and typographical errors throughout the paper.

Reviewer 2 Report
Comments and Suggestions for Authors
The main question addressed by the paper is the possibility to apply more methods to improve human-computer interaction for disabled persons. The paper proposes and presents a 3-modal human-computer interaction (3M-HCI) system. It is a novel interaction system that dynamically integrates 3 inputs (facial, vocal, and eye-based inputs) through a new signal processing pipeline and a cross-modal coordination mechanism.
The topic is not entirely original, but it is very relevant to the field and gives very interesting and good results. It addresses a specific area in the field that has not been sufficiently researched and described. So, there is great interest in such innovative approaches and proposals for solving the problems of disabled people when working with computers, or interacting with computers.
This approach adds some advantages compared with other published methods. It not only enhances recognition accuracy but also reduces interaction latency. Experimental results are very interesting and demonstrate that this method outperforms several recent hands-free interaction solutions in both speed and precision, highlighting its potential as a robust assistive interface.
It would be needed to more consider possible disadvantages of such an approach, as well as problems in the implementation and application. Also, it would be good to more emphasize what are the advantages and disadvantages compared to some other methods known from the literature.
In the conclusions it would be good to consider and emphasize again the advantages and also the disadvantages of such an approach, and problems in the implementation and application.
The references are appropriate.
The images should be mentioned first in the text and then to appear in the paper. This is needed to correct (Figure 1, Figure 12). Figure 6 is not mentioned in the text at all.
Comments on the Quality of English Language
Used English language and grammar could be improved.
Author Response
Comments 1: It would be needed to more consider possible disadvantages of such an approach, as well as problems in the implementation and application. Also, it would be good to more emphasize what are the advantages and disadvantages compared to some other methods known from the literature.
In the conclusions it would be good to consider and emphasize again the advantages and also the disadvantages of such an approach, and problems in the implementation and application.
Thank you for the comment. We have added a more detailed discussion in the Limitations section, highlighting possible disadvantages of our approach, implementation challenges, and a comparison of advantages and disadvantages relative to other methods reported in the literature. In addition, we have briefly emphasized these advantages and disadvantages again in the Conclusions.
Comments 2: The images should be mentioned first in the text and then to appear in the paper. This is needed to correct (Figure 1, Figure 12). Figure 6 is not mentioned in the text at all.
We have revised the manuscript to ensure that all figures are mentioned in the text before they appear, and Figure 6 has now been properly referenced in the text.

Reviewer 3 Report
Comments and Suggestions for Authors
Authors present a 3M-HCI model for hand-free computer interaction. By utilizing openCV, mediapipe, dragonfly2 with customtktinter as interface builder and pyautogui and numpy as computation libraries, main contribution is at coordination among these input toolkits and a process logic for inputs.
However, the system architecture is not clearly indicated these contributions (Figure 1 and 5) and modules that authors have used such as openCV, mediapipe and dragonfly2. Although GUI and mouse accessibility is not the focus, the keys for integrating these three toolkits should be clearly described and technology sound should be evaluated.
FIgure 1 is not a system architecture, should be redrawn. Figure 5 is only a voice input process diagram and should be renamed and provide a process diagram indicating these three types of inputs. A system architecture should be provided and clearly describe the technical reasons.
Coordination among these three modules (facial, voices and eye-gaze) need to further be defined and evaluated.
Author Response
Comments 1: “Authors present a 3M-HCI model for hand-free computer interaction. By utilizing openCV, mediapipe, dragonfly2 with customtktinter as interface builder and pyautogui and numpy as computation libraries, main contribution is at coordination among these input toolkits and a process logic for inputs.”
We agree that one of our key contributions lies in incorporating cross-modal coordination among input modalities to enhance the hands-free user experience. At the same time, we would like to emphasize that our contributions are not limited to the integration aspect. In addition to the cross-modal coordination across different input modalities, we also introduce:
- an adaptive filtering mechanism that effectively suppresses signal noise while maintaining low-latency responsiveness
- a redefined mapping strategy from input signals to cursor movement to achieve more precise control.
The Introduction section has been revised to present these contributions more clearly
Comments 2: “However, the system architecture is not clearly indicated these contributions (Figure 1 and 5) and modules that authors have used such as openCV, mediapipe and dragonfly2. Although GUI and mouse accessibility is not the focus, the keys for integrating these three toolkits should be clearly described and technology sound should be evaluated.”
We thank you for the valuable comment. The System Architecture section has been revised to provide a clearer overview of the employed modules and to highlight the key aspects of toolkit integration.
- OpenCV is adopted to capture and processing image because it is implemented in optimized C/C++, delivers high performance and a comprehensive set of tools for efficient real-time image processing (line 141).
- MediaPipe was selected for face processing after a careful comparison with other toolkits. The detailed comparison is provided in Section 3.1.1 (line 182).
- Since voice command is not the primary focus of our system, we chose Microsoft’s Speech API (SAPI5) as it offers an easy-to-integrate, lightweight solution that meets our requirements, ensuring consistent performance, low latency, and minimal hardware demand.
Comments 3:“Figure 1 is not a system architecture, should be redrawn. Figure 5 is only a voice input process diagram and should be renamed and provide a process diagram indicating these three types of inputs. A system architecture should be provided and clearly describe the technical reasons.”
We have carefully redesigned Figure 1 to present a clear system architecture, and we have expanded the corresponding section to provide a detailed explanation of the technical rationale behind it. In addition, Figure 5 has been revised to illustrate the complete input processing pipeline, encompassing all three input modalities rather than only voice input. This updated diagram is placed within the general system architecture section, with further details provided in the respective subsections.
Comments 4: “Coordination among these three modules (facial, voice and eye-gaze) need to further be defined and evaluated.”
Thank you for your constructive feedback, we have enhanced clarity of cross-modal coordination in System Architecture (line 145). In general, the face processor validates input for the voice processor, while the voice processor can adjust profile settings and influence the behavior of the face processor. Both modules interface with the operating system through PyAutoGUI, which enables seamless control of mouse and keyboard actions. In our system, eye-gaze is modeled as part of facial expressions, enabling intuitive and fine-grained interaction with the system via PyAutoGUI. The effectiveness of these mechanisms can be verified through our released project code.

Round 2
Reviewer 3 Report
Comments and Suggestions for Authors
Dear Authors,
thank you for refining the manuscript to reflect my comments.
Current version is much better than previous and clarify several issues I mentioned. I have only one comment:
In order to clarify the contribution as declared, there should be a clear block indicated the contribution such as the filter in addition to the integration part. Ideally the components that authors have developed should be clear indicated and described in addition to the open source components (such as opencv and so on).
Author Response
Comments: In order to clarify the contribution as declared, there should be a clear block indicated the contribution such as the filter in addition to the integration part. Ideally the components that authors have developed should be clear indicated and described in addition to the open source components (such as opencv and so on).
Thank you very much for your valuable comments to improve our manuscript. In order to clarify our contribution, we have revised the Figure 1 and Figure 2 of the manuscripts. These revisions help to clearly indicate the new idea of the authors.
